# Urban Pest Abundance and Public Enquiries in Zurich 1991–2022

**DOI:** 10.3390/insects14100798

**Published:** 2023-10-02

**Authors:** Peter Brimblecombe, Gabi Müller, Marcus Schmidt, Werner Tischhauser, Isabelle Landau, Pascal Querner

**Affiliations:** 1Department of Marine Environment and Engineering, National Sun Yat-sen University, Kaohsiung 80424, Taiwan; p.brimblecombe@uea.ac.uk; 2School of Environmental Sciences, University of East Anglia, Norwich NR4 7TJ, UK; 3City of Zurich, Department of Environment and Public Health, Urban Pest Advisory Service, Eggbühlstrasse 23, CH-8050 Zurich, Switzerland; gabi.mueller@zuerich.ch (G.M.); marcus.schmidt@zuerich.ch (M.S.); werner.tischhauser@zuerich.ch (W.T.); isabelle.landau@zuerich.ch (I.L.); 4Natural History Museum Vienna, Burgring 7, 1010 Vienna, Austria; 5Department of Integrative Biology and Biodiversity Research, Institute of Zoology, University of Natural Resources and Life Sciences, Gregor-Mendel-Straße 33, 1180 Vienna, Austria

**Keywords:** awareness campaigns, climate change, COVID-19, media reporting, pest control

## Abstract

**Simple Summary:**

Zurich’s Urban Pest Advisory Service (UPAS) gives the public the opportunity to report household pests and join attempts to reduce potential health threats. More than fifty thousand records, dating back to the 1990s, provide information on the changing prevalence of pest problems in Zurich. Cockroaches declined most notably over this period, possibly through the availability of effective gel baits. From 2010, there was much interest in bed bugs and awareness campaigns made travellers aware of how to avoid bringing bed bugs home after summer holidays abroad. In recent years the Asian tiger mosquito has been found in Switzerland and has raised much public interest, though few were identified in Zurich. During COVID-19 people spent more time at home, which may have led to more observations of wasps.

**Abstract:**

Zurich’s Urban Pest Advisory Service (UPAS) aims to survey, control and reduce hazards posed by pests. Records submitted to the UPAS may not exactly correlate with abundance but can reveal patterns of change. These reflect changes in species, public and media perceptions and the effects of climate and COVID-19, along with the effectiveness of new pest controls. Records for *Blattodea* and *Plodia interpunctella* declined in the period 1990–2022, while *Cimex lectularius* and *Psocoptera* increased. Summer has typically revealed the largest number of insects reported and Google searches show parallel seasonal variations. The *Blattodea* declined five-fold over time, likely due to better pest control methods (gel baits). *Aedes albopictus*, though rare, was the subject of media reports and awareness campaigns, which resulted in much public interest. Vespidae are abundant and have been reported in sufficient numbers to warrant an analysis of seasonal records, suggesting that July temperatures affect numbers. COVID-19 restrictions led to more frequent reports of rodents, pigeons, *Zygentoma* and *Stegobium paniceum*. The long-term reporting to the UPAS gives a useful indication of the changing concerns about pests in Zurich.

## 1. Introduction

Insects, rodents and birds are common pests, nuisances in cities, and are a source of hygiene problems, food loss and material damage, in addition to being disease vectors. However, evaluating the most effective control method for pests can be a difficult problem. Large sums of money are spent across the globe on urban pest control in the food industry, restaurants, homes and offices—the global pest control market was valued at $22.6 billion in 2022 [1]. Household pests in cities are rarely monitored in a systematic way as pest control companies treat a large number of infestations but do not typically analyse or publish the relevant data.

The Urban Pest Advisory Service (UPAS), in Zurich, aims to survey, control and eliminate health hazards posed by insects, other arthropods that cause nuisance, and rodents [2]. This unique agency is the only non-commercial and official urban pest advisory service in Switzerland and has employees who have an education in entomology and in pest control [3]. It focuses on urban pests, and not, for example, agricultural pests, or human, animal and plant pathogens. The UPAS took on its current form with the appointment of two biologists and an agronomist in the 1990s and is integrated within the Environment and Public Health Department of the City of Zurich. 

Since the 1990s the enquiries and insect samples sent by members of the public to the UPAS have been documented in a digital database. We used this dataset to analyse trends and changes in the reporting of pests, mainly insects, in the city of Zürich for the period 1991–2022, avoiding incomplete years 1990 and 2023. Some of these records are questions from the public and enquiries from journalists. Further submissions come from pest control companies working in public buildings belonging to the City of Zurich. The pest control companies are now required to make these submissions, as specimens, under the public procurement laws. The records submitted to the UPAS show changes over time [2,3]. This current study attempts to assess the main drivers of temporal change in the database and extensive pest reporting of individual species. Because of the regular features in the media (newspapers, radio and TV) on pest activity and new species, we want to determine whether more pests are reported after media activity or if such activity is a response to an increase in pest activity. Further effects of climate, COVID-19 lockdowns and changing social behaviour are also possible influences on pest abundance, activity and reports and queries from the public. Additionally, long term changes to the management and control of pests (such as new and more efficient products to control pests) can directly influence the abundance of individual species in the city.

## 2. Materials and Methods

### 2.1. Data Collection and Storage

Data collection by the UPAS has been primarily designed to support its official activities, with reports and enquiries to the UPAS stored in Microsoft Dynamics 365, and has similarities with the collection of data within medical entomology departments [4], historic houses [5,6] or museums [7]. Often, such institutions use Microsoft Excel to handle data, and though such collections of data are valuable in research, they need to be tuned to the needs of studies [8]. The UPAS records capture the exact address (street name and number) and dates of occurrence, the species names or animal groups of the pests (as far as can be verified), the name of the reporting person (optional) and specific remarks on the case. The database contains a specially designed list of all reported animals (including pests) based on the scientific systematics (phylum, class, order, family, genus, species) that, according to Sauter [9], Stresemann et al. [10] and Weidner [11], are typically used for classification of species. For each systematic group there is a 3-digit code which ultimately forms part of a larger, 18 digit, code for a species. Often, people report animals to family or do so only at the group level (e.g., mice, pigeons, ants or wasps). Unless they also send samples for identification by an expert of the UPAS, the pests can only be registered in terms of higher taxonomic level—the public may easily misidentify them. 

The data gathered between 1990 and spring 2023 includes more than 57,000 individual data recordings, although we have avoided including the incomplete years at the beginning and end in the analysis. This was downloaded as a comma separated variable file (csv) and explored with short scripts written in gawk [12]. We were able to separate the media requests for information about insects and submissions from pest control companies. This was useful in the exploration of journalistic activities and of new regulations for reporting pest eradication efforts in public buildings. Our analysis was simplified by focusing on the more dominant species or groups of insects: (i) cockroaches (*Blattodea*), including genera: *Ectobius*, *Blattella*, *Supella* and *Periplaneta;* (ii) wasps and hornets (*Vespidae*); (iii) ants (*Formicidae*); including *Lasius* spp. (but not the imported species *Monomorium pharaonis* Linnaeus, 1758 and *Lasius neglectus* Van Loon, Boomsma & Andrásfalvy, 1990); (iv) the Indian meal moth (*Plodia interpunctella* Hübner, 1813, family: *Pyralidae*); (v) bed bug (*Cimex lectularius* Linnaeus, 1758, family: *Cimicidae*); (vi) silverfish (*Zygentoma*, including *Lepisma saccharinum* Linnaeus, 1758 the common silverfish, *Ctenolepisma longicaudatum* Escherich, 1905 grey silverfish, *C. lineatum* Fabricius, 1775 the four-lined silverfish and *C. calvum* Ritter, 1910 the ghost silverfish) and (vii) the booklice (*Psocoptera*). Occasionally we introduce other species for comparison. 

We analysed the changes over time and season to explore the effect on changing control method such as bait gels, median and awareness campaigns, climate, seasonality and mobility under COVID-19 restrictions. 

### 2.2. Internet Searches for Pests and Treatments

We used Google Trends (trends.google.com (accessed on 20 July 2023)) to assess the relative volume of searches for a given term. This has been used, for example, in medical science to assess the number of searches related to tick-borne encephalitis in Germany. In this instance these numbers reveal a clear seasonal pattern with warm season highs, though there was no relationship between case numbers and searches [13]. This gives a normalised output such that the maximum month is 100. The data can be tuned to location, so we used Switzerland, with most searches coming from Zurich. The search volume is available from 2004, although there have been some changes to geographical assignment (in 2011) and data collection was improved in 2016 and 2022, so care is required for these time thresholds.

### 2.3. Climate Data and COVID-19 Pandemic

Climate observations came from the meteorological station of MeteoSwiss (https://www.meteoswiss.admin.ch (accessed on 15 July 2023)) focusing on the record from Zürich–Fluntern at an elevation of 556 m.

The study includes the years of the COVID-19 pandemic, so the restrictions over this time may have potentially affected organisms and records in the database. The Federal Council of Switzerland started to limit public gatherings in late February 2020 in order to restrict the spread of the virus. There was no complete lockdown, but from March authorities increasingly tried to protect the most vulnerable individuals and later closed bars, shops, schools and other gathering places. Restrictions eased from late April to early June, but in October 2020, following a rapid increase in cases, stricter measures were imposed limiting public gatherings and encouraging working from home. In January 2021, after a month in which cases remained at a high level, additional measures were taken. By September 2021 the vaccinated and the population previously infected with COVID-19 were able to access most facilities. In February 2022, almost all measures were lifted.

### 2.4. Statistical Analysis

As the data were not always normally distributed, we used non-parametric tests. Trends were typically manifest as Theil–Sen slopes [14], which can be considered the median of the slopes of all lines through pairs of points and are robust against outliers. We used the Kendall rank correlation (statistic *τ*) as a measure of association between two measured series and which works in a similar manner to a normal Pearson regression coefficient (*r*). Although statistical analysis was used to assess the probability of changes in the suggested data, it offers less guidance on the causes of change.

## 3. Results and Discussion

As more than 2000 reports and enquiries are received at the UPAS each year the collection represents a significant amount of information about pests in the city of Zürich. However, the record consists of a range of material from inquiries and observations of insects by the general public through to the submission of samples. This heterogeneous input of records to the database means that it is a biased sample, but one which potentially reflects the abundance of the most common species. 

Nevertheless, there are structures within the changes in the record that give an indication of important drivers that influence reporting to the UPAS. The *Blattodea* have been in decline; from more than 500 reports each year 30 years ago to barely 100 at the present time. Individual species declined over different times, with more effective control measures driving these declines. The same was likely true of *P. interpunctella*, which was well recognized in households and often introduced as eggs that were found in dried food or animal food (bird or rodent food) that had been stored too long. 

### 3.1. Trends in Pest Reporting

The database of pest reports and enquiries to the UPAS began in 1990 and continues to run through to the present, although it is fragmentary in its first year. Initially these reports were recorded on data sheets, but these were transferred into Microsoft Access in 1997, and were thereafter entered in real time. As of early 2023, there were more than 57,000 records, with the majority from the city of Zürich (41,726), 527 records from Winterthur (a city in the canton of Zürich), 253 records from Basel in north-western Switzerland, smaller numbers of records from other areas that include places within the canton of Zürich (e.g., Adliswil, Uster, etc.) and 1561 records where no location is specified. The database includes a wide variety of pests, and although most are arthropods, there are records of other pests, e.g., rats (mostly *Rattus norvegicus* Berkenhout, 1769) and mice (especially the house mouse, *Mus musculus* Linnaeus, 1758) and pigeons (*Columbiformes*). 

Figure 1a shows the total number of records for the seven most abundant groups of insects: (i) *Blattodea*, (ii) *Vespidae*, (iii) *Formicidae*, (iv) *P. interpunctella*, (v) *C. lectularius*, (vi) *Zygentoma* and (vii) *Psocoptera*. As shown in Figure 1b the total number of records has been fairly constant over the years (except for 1990 when there were only 115). The constant number means that there is a lack of changes in reporting levels that are large enough to have substantially affected the year-to-year change in the representation of a particular insect species (Figure 1c–i). 

The change in the number of records over time can be seen to vary between the abundant groups of insects. The Theil–Sen slopes are significant (*p* < 0.0001) and negative for the *Blattodea* and *P. interpunctella*. In the late 1990s the numbers of *Blattella germanica* Linnaeus, 1767 (German cockroach) declined, while there was an increase in reports of *Ectobius vittiventris* Costa, 1847 (amber wood cockroach). *E. vittiventris* is a typical outdoor insect that was initially confused with *B. germanica* by the public and many pest control companies [2]. After some years, *E. vittiventris* saw declining numbers of reports as the public began to learn to distinguish this species, emerging from their gardens and outside habitats, from the similar looking hygiene pest *B. germanica* (inset to Figure 1c). The occurrence and coordinated control of the oriental cockroach (*Blatta orientalis* Linnaeus, 1758) in Zurich has been described by Landau et al. [15]. 

In contrast with the *Blattodea* and *P. interpunctella*, *C. lectularius* and the *Psocoptera* both show significant (*p* < 0.0001 and *p*~0.0002) increases. Although the *Formicidae* show elevated numbers in recent years, the slope is not significant (*p*~0.65). Focusing on the most frequently recorded pests means that some important insect types may be missed. In particular, those that damage fabric and furnishings are typically reported less than a thousand times, e.g., the webbing clothes moth (*Tineola bisselliella* Hummel, 1823), carpet beetles (Genus: *Attagenus* Latreille, 1802; Genus: *Anthrenus* Geoffroy, 1762) and food insects such as the biscuit beetle (*Stegobium paniceum* Linnaeus, 1758). Additionally, there are some invasive species that have been found in Zurich in recent years, such as the tiger mosquito (*Aedes albopictus* Skuse, 1894), brown marmorated stink bug (*Halyomorpha halys* Stål, 1855), western conifer seed bug (*Leptoglossus occidentalis* Heidemann, 1910) and the pharaoh ant (*Monomorium pharaonis* Linnaeus, 1758). 

*Aedes albopictus* is rare in most years, with numbers of the species caught over the last decade being in the single digits, excepting 2019 when more than 500 were suspected but in which just 3 occurrences were ultimately found, 2 in the city of Zurich and 1 in Horgen (in the Canton of Zurich). Nevertheless, the mosquito is worrisome as a vector of diseases ranging from mild flu-like illness to severe dengue fever [16] and there is some concern that its range in Switzerland will expand in the second half of the 21st century [17], with Zurich a particular target [18]. *H. halys* rose rapidly at the end of the first decade of the 21st century to 91 records for 2010, after which the number of records about the insect gradually declined. The number of records of *M. pharaonis* has grown slowly over the years with a positive slope (*p*~0.02) and around 12 per annum over the last decade.

### 3.2. Seasonal Change

There was a consistent cycle of public reporting of the *Vespidae* with a peak in the early summer, which corresponded with the seasonal searches for “Wespe” on Google. This corresponds to a time when wasp colonies are likely to be fully populated and the insects are most active around their nests. Public interest is also driven by media attention and awareness campaigns. Early summer press releases from the UPAS about bed bug (*C. lectularius*) prevention were likely responsible for media interest and a high frequency of reports from the public for a number of years. Rare insects such as *Ae. albopictus* were the subject of media attention and leafleting by the UPAS. This led to many reports of suspected observations of *Ae. albopictus* to the UPAS, though only in very few cases was it found to be truly *Ae. albopictus* after verification by the UPAS. This is clearly a case where the number of records in the database reflect interest in the insect rather than its abundance.

Wasps are among the most reported insects to the UPAS, with some 8000 records. The European hornet (*Vespa crabro* Linnaeus, 1758) was most often (1117) identified at species level; additionally, the common wasp (*Vespula vulgaris* Linnaeus, 1758) and the German yellowjacket (*Vespula germanica* Fabricius, 1793) were found to have 210 and 246 records, respectively, as well as *Polistes dominulus* Christ, 1791 with 256 records, and *Dolichovespula saxonica* Fabricius, 1793 with 298 records. In our initial analysis all of the *Vespidae* were grouped together. As the records for these insects did not show significant trends over time, we were able to explore the trend in the monthly records from 1991. Figure 2a illustrates a regular annual pattern, with the largest number of records submitted mid-year. The cycle can also be seen in the relative number of Google searches for “Wespe” (German for wasp) in Switzerland since 2004. There is a similarity in the seasonal pattern that provides evidence for a consistent cycle of public attention to wasps, which are likely to be evident in the early summer when wasp colonies are likely to be fully populated and the insects most likely to fly out from their nests. There is a significant correlation (Kendall *τ* = 0.37 and *p* < 0.00001) between the monthly number of Google searches and the number of records of wasps to the UPAS. 

The pattern of high reporting in the summer is repeated for all key groups of abundant insects reported to the UPAS (Figure 2b–i). However, the shapes of the distributions throughout the year differ among the insects. It is narrow for the outdoor insects e.g., *Ectobius* and the *Vespidae*, but rather broad for indoor insects such as *P. interpunctella*. As an example, the *Formicidae* are often a problem in spring, when they find insufficient food outside, enter houses and are thus seen as a pest and reported as such to the UPAS. The peak is late in the year, October for the *Zygentoma* and November for the *Psocoptera*. However, *T. bisselliella* Hummel, 1823 (webbing clothes moth) peaks in May. These are not necessarily the same as the catches in museums and libraries, where the *Zygentoma*, for example, tend to peak rather early [19]. The summer peak of *C. lectularius* could result from the pests coming back with holidaymakers’ luggage [20,21].

### 3.3. Google Searches

Although the seasonal pattern of Google searches for insect pests matched our expectations, with higher frequency in the summer, longer term changes were not clear, e.g., increasing searches for ants did not agree with reports to the UPAS, which were relatively stable. While it might be argued that households sought internet advice to deal with the insects, there was no rise in the searches for ant control.

Figure 2a has shown that there is a relationship between the number of Google Searches for “Wespe” and records collected by the UPAS. Figure 3a shows the changes in the relative numbers of searches each year for “Schabe” (*Blattodea*), “Wespe” (*Vespidae*), “Ameise” (*Formicidae*), “Dörrobstmotte” (*P. interpunctella*), “Bettwanze” (*C. lectularius*) and “Silberfischchen” (*Zygentoma*). These are not necessarily an accurate reflection of searches for a particular insect as it can be misidentified by the public, e.g., even in the tropics, where bed bugs are a well-known problem, the public can often misidentify *C. lectularius* [22,23].

The trends in the data suggest that searches for “Schabe” and “Silberfischchen” are in decline (*p* < 0.1) or stable for Dörrobstmotte and Bettwanze. Two search terms increase over time (*p* < 0.005): “Wespe” and “Ameise”. The increase in the search volume for the *Formicidae* is particularly notable, but enquiries to the UPAS have been relatively stable, except for the COVID-19 years, 2020–2022. The Google search volume for “Pharaoameise” has declined (Figure 3b) as it has for searches using “Ameisen bekämpfen” (i.e., controlling ants). The latter term captures searches by people who want to eradicate ants by themselves in addition to those who are looking for a pest control company. 

### 3.4. Journalism and Awareness Campaigns

Long term media interest and awareness campaigns about bed bugs increased the number of records in the UPAS database. Although there was a relationship between records of *Vespidae* in the UPAS database and media stories there is no evidence they were stimulating public interest in the case of this well-known insect. In contrast, it may be that, with *C. lectularius*, and even more so with *Ae. Albopictus*, media coverage stimulated public interest in these invasive species.

As many of the records in the UPAS database come from the public, the numbers are sensitive to public perception. This is readily influenced by a number of factors other than the simple abundance of a pest. The UPAS received almost 370 enquiries from the media between 1990–2022. Numbers were relatively small in the first 10 years, with just 8 enquiries. However, since 2000 the number of enquiries has risen and in the last 20 years has settled at about 17 ± 5.7 per year. As shown in Figure 4a most media enquiries concern *Vespidae*, while most reports to the UPAS concern *Blattodea*. Thus, media enquiries do not seem to reflect pest occurrence. The media is also surprisingly active in enquiring about *Rodentia* (*Rattus* sp. and *Mus* sp.), *C. lectularius* and *Ae. albopictus* compared with the records in the UPAS database (Figure 4a). This might be the result of media releases by the UPAS to raise awareness about these pests. Nevertheless, there were also some enquiries from the media about the *Formicidae* and *P. interpunctella*, which may cause problems in kitchens. However, there was only modest interest in the *Zygentoma* or *T. bisselliella*. There were a number of media enquiries about *Rhinotermitidae or Termitidae* and these may reflect increasing concern with these species, though reports on *Rhinotermitidae* are found only once in the UPAS records. There was no media interest in the *Psocoptera* which is in line with the argument of Bertone et al. [24] and Wang et al. [25] and may be due to a prevalence in storage areas [26]. As noted previously, *H. halys* was prevalent early in the 21st century, though records declined after a high of 91 records in 2010. However, media interest, with enquiries in 2009, 2011, 2012, 2013 (4), 2018 (2) and 2019, has seemed more persistent than observations of the species itself, perhaps a result of its pungent odour. 

Examining the *Vespidae* more closely is possible, as there are a large number of media enquiries that peak in July and August (Figure 4b), in line with the warmer season. This is a similar pattern to the number of records in the UPAS database (Figure 2c). The number of UPAS reports (June–August) shows a weak correlation with the media enquiries (July–August), as seen in Figure 4c. The reports to the UPAS seem to become frequent slightly earlier than articles in the media, so it is likely that the media enquiries are in response to the elevated level of public interest. 

Public awareness is understandably enhanced by information campaigns, most notably for issues with *C. lectularius* (bed bugs). The UPAS produced press releases about prevention before the start of the summer holidays in the years 2011, 2013, 2015, 2016 and 2022 (Figure 5a), distributing information on how to avoid bringing bed bugs home when returning from holiday. These press releases aligned with growing numbers of reports to the UPAS (peaking in 2016), and continued media attention over earlier years. The UPAS undertook a survey with Swiss Pest Control Companies in 2017 assessing the cases requiring the control of bed bugs across the years 2011–2017. Unfortunately, it is difficult to obtain returns to these surveys because of an understandable commercial sensitivity, so only six replies were obtained from more than 50 companies in the Swiss Pest Control Association. This reflected a reluctance to reveal methods or properties treated. 

The UPAS began an awareness campaign about *Ae. albopictus* in 2019, after the mosquito was found in a residential area. This was distributed in the form of a flyer included in a letter to house owners and residents in Zurich’s District 2 and was followed with a press release and an information evening on 20 August 2019 for residents, house owners and the media. A control zone of 5 hectares was created, a narrow area some 500 m in length, where mosquito larvae were controlled in sewer holes and house owners were asked to eliminate breeding sites from their gardens. A larger monitoring zone was assessed using traps. Google searches for “Tigermücke” showed an annual cycle of interest with summer maxima that reached a peak at the same time as reports to the UPAS (Figure 5b). In the case of this invasive insect, newspaper articles and radio broadcasts preceded public interest. However, there were many enquiries of suspected *Ae. albopictus* (516) in 2019, but these were largely misidentification by the public. After the successful eradication of this small local *Ae. albopictus* population in Zurich Wollishofen, interest dropped in the following year, 2020 (278), but rose again as *Ae. albopictus* began to establish in other Swiss cantons (Basel Stadt, Basel Land, Geneva, Vaud, Valais) and resulted in more public reports of concern about *Ae. albopitus* in 2021 (502) and 2022 (531).

### 3.5. Climate Effects

It is often argued that the year-to-year variation in population is a function of climate, but this is not always readily evident given the range of environmental factors that affect the development of populations and social factors that influence the reporting of insects. However, the *Vespidae* are easily recognised and frequently recorded over the years so are a useful pest to examine for the effects of climate. Here, it seemed that the frequency of records in the summer (July–August) were a function of the July temperature, seemingly a climate effect. Climate change is also seen as an important control on insect populations. This can be an even more subtle change over the long term, but again the record of the *Vespidae* is large enough to show that reports are arising earlier in the year, consistent with a warmer climate. Such evidence of weather effects was more difficult to discern in the database with other species. However, over time the warmer summers may increase and so may make changes more obvious. 

There are many studies of the effect of climate on urban insects as reviewed by Dhang [27]. The extensive record and stable population of the *Vespidae* mean that they can be effectively studied via the UPAS database. *Vespidae* have shown the importance of temperature regulation in wasps [28], although it is both temperature and pressure that seem to regulate the phenology of wasps in Poland [29]. The UPAS data show the number of records for *Vespidae* are at their highest in July and August, the sum across these two months (*N*_7–8_) was compared with the monthly temperature (*T*) and precipitation (*P*) for the years 1991–2022, giving the best fit to the multiple regression equation:*N*_7–8_ = 34 + 2.9*T*_4_ + 2.2*T*_5_ − **27.9***T*_6_ + **23.4***T*_7_ + 8.1*T*_8_ − 0.12*P*_4_ − 0.20*P*_5_ − 0.49*P*_6_ + 0.28*P*_7_ − 0.04*P*_8_(1)
where only the temperature parameters for June and July (bold face with *T*_6_ and *T*_7_) are significant at the *p*_2_~0.1 level, with an *R*^2^ of 0.415. The trend in the number of July–August records over the years are compared with the number predicted by Equation (1) in Figure 6a as a concordance that becomes more obvious if the quadratic trend is removed, as seen in the inset. 

Multiple regression suggested that July temperatures were most important in explaining the number of reports concerning *Vespidae* each July–August, the correlation was only partial, as shown in Figure 6b (Kendall *τ* = 0.3, *p*_2_~0.018). The multivariate analysis was repeated for the *Formicidae* over the peak reporting months April–July, but there was no significant correlation (i.e., *p*_2_ > 0.1 with *R*^2^ of 0.13) for any of the monthly average temperatures or rainfall amounts; though there were hints of a slight positive relationship to June temperatures (i.e., *p*_2_ ~> 0.13). In the case of *Cimex* sp., which peaks in August and September, the numbers were weakly correlated (*R*^2^~0.27) in a multiple regression with monthly temperatures (May–September), with June temperatures showing a significant positive correlation (*p*_2_ = 0.06). 

It is also possible to assess the more continuous change in climate by looking at changing seasonal patterns brought about by rising temperatures. We might expect the increase in reports to arise slightly earlier and to perhaps show a broader spread as previously found for the phenological characteristics of *Vespidae* [29]. Figure 7 hints that peak reporting numbers arrive earlier for the *Vespidae*, and arguably for *P. interpunctella*. In the case of *Formicidae* and *Zygentoma*, there is scant evidence of any change over time. *P. interpunctella* and *Zygentoma* both live mainly indoors and are therefore less influenced by the outdoor climate.

### 3.6. COVID-19 Effects

COVID-19 was present in Switzerland from early 2020 and from late February 2020 public gatherings were increasingly restricted, leading to a dramatic shift in the places where people spent their time. The pandemic appears to have made some pests, such as *Rodentia*, *Zygentoma*, *S. paniceum* and *Columbidae* more apparent, as a part of the changing observations of urban fauna [19,30,31,32]. The year 2020 saw the largest ever number of records (2369) in the UPAS database, compared with the year before (2019: 2267) and after (2021: 1922). However, part of the high numbers found in 2020 arose as submissions from Pest Control Companies undertaking pest control in civic properties. Nevertheless, these submissions emphasised a rising problem with mice, that had been an infrequent source of complaint in earlier years (Table 1). This could arise because people were at home more or because pests were ranging more widely. 

Not all pests increased during the COVID-19 years 2020–2021, although numbers for *Vespidae*, *Formicidae*, *Zygentoma*, *Attagenus*, *Stegobium*, *Columbidae* and *Monomorium* spp. are shown to be high under the restrictions. However, the Friedman test for the 15 pests examined shows that there is a tendency for 2020/2021 to be higher than average for 2018/2019 and 2022. This applies to the following 15 pests: (i) *Rattus*, (ii) *Mus*, (iii) *Blattodea*, (iv) *Vespidae*, (v) *Zygentoma*, (vi) *Formicidae*, (vii) *C. lectularius*, (viii) *P. interpunctella*, (ix) *Psocoptera*, (x) *Anthrenus* spp., (xi) *Attagenus* spp., (xii) *S. paniceum*, (xiii) *Columbidae*, (xiv) *Halyomorpha* and (xiv) *Monomorium* spp. The mean ranks for the reported annual catch are 2.1, 2.4 and 1.5 for the periods before, during and after the pandemic restrictions, 2018–2019, 2020–2021 and 2022, respectively, and are moderately significant at *p* = 0.067. This is not strongly significant, but one insect, *Anthrenus* sp., was at its lowest during the COVD-19 restrictions. *Cimex* spp. and *Psocoptera* declined across the period, while *Rattus* spp., *Mus* spp. and *Columbidae* tend to be higher even after the pandemic restrictions ended.

Some of the increases in reporting during COVID-19 may have been caused by people spending more time at home. This might be true for the *Vespidae* as there were increased searches for “Wespe” on the internet in 2020 (Figure 2a) which corresponded with reports to the UPAS which rose steeply in the spring of that year, a time when the queens search for suitable places to build their nests. However, it may be true that this was more noticeable when fewer people were at work, and it was also an unusually dry and sunny spring for Switzerland when Zurich temperatures in April were some 4 °C higher than normal [33]. 

### 3.7. Summary Outcomes

This study deals with reports submitted to the UPAS, but these observations of trapped pests were not always verified by experts. Where such confirmation was not possible, reports were recorded as the order, family or genera in order to ensure false information was not added into the data base. However, when a species was recorded, a specimen (or more) was delivered to the UPAS and identified by an expert. Bias in self-reported data is a familiar problem in epidemiology [34,35] typically relating to recall issues where time has passed, or where there is assessment fatigue and social desirability bias [36]. These issues are not especially relevant to the UPAS records, although it is possible that there might be some desirability bias in the under reporting of pests that convey a negative image (e.g., poor housekeeping). 

There was an evident decline in reports of *B. germanica* in the 1990s, although somewhat delayed in the case of the outdoor cockroach *E. vittiventris*. Indoor cockroaches were effectively controlled by gel baits [2]. The decline in reports of *E. vittiventris* was likely a product of an increasing realisation that the insect was an accidental visitor to the garden rather than a hygiene pest. *Cimex lectularius* became an important European problem at the beginning of the 21st century [20,21] and reports to the UPAS were typically made in summer and autumn, arising from holidaymakers who returned home with bites or pests in their luggage.

There was often a discrepancy between Google internet searchers and reports to the UPAS. The increasing search volume for the *Formicidae* is particularly notable as a time when ants were reported less frequently, although the invasive ant, *M. pharaonis*, showed a slight increase in the number of reports (*p*~0.02) over the study period. In general, Google search volume appears poorly correlated with reports to the UPAS, although it follows the expected seasonality with warm season increases. 

We found no media interest in the *Psocoptera*, which is in line with the argument of Bertone et al. [24] and Wang et al. [25] that the insect is small and often unnoticed. Despite this, booklice and bark flies are frequent visitors in the domestic setting and there are many hundreds of records in the UPAS database; additionally, they are common in museum storage areas [26] and historic houses [5]. The UPAS often receives reports of booklice from new buildings where the cement structures were not properly dried before floors were inserted and occupants moved in. They are also a common problem in many housing estates with higher indoor humidity.

COVID-19 presented more opportunity for households to observe insects as lockdowns and other restrictions meant that people spent more time in their homes. This seems a likely explanation for an increased level of reporting for some pests most notably *Rodentia*, *Zygentoma*, *S. paniceum* and *Columbidae* during COVID-19 restrictions.

## 4. Conclusions

The reports to the UPAS, while influenced by public and media perceptions, nevertheless carry important information about the changes that are presently underway. In addition to the social drivers of the changes in reporting, there are indications of the influence of environmental factors, such as climate and climate change. This study has often focussed on groups of insects rather than looking at changes at the species level within a genus or family, so further studies should examine that. We have still to undertake surveys of the public experience of urban pests and explore the ways in which analysis of the data contributes to better management of urban pests. However, this initial analysis shows that the database, despite its heterogeneity and an administrative origin, carries valuable information for further research. 

## Figures and Tables

**Figure 1 insects-14-00798-f001:**
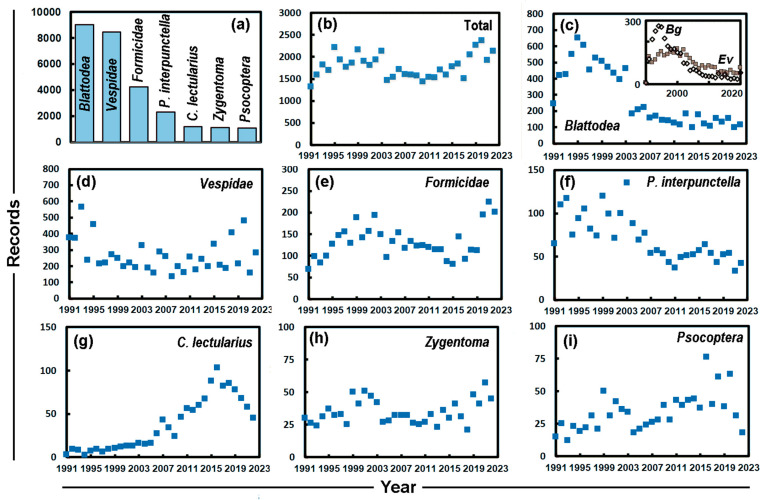
(**a**) Bar chart showing major groups of insects reported. Trends 1991–2022 for: (**b**) total number records each year; (**c**) cockroaches (*Blattodea)*, where the inset shows *Blattella germanica*, *Bg* and *Ectobius vittiventris*, *Ev*; (**d**) wasps and hornets (*Vespidae)*; (**e**) ants (*Formicidae)*; (**f**) the Indian meal moth, *P. interpunctella*; (**g**) the bed bug (*C. lectularius)*; (**h**) silverfish (*Zygentoma*); and (**i**) booklice (*Psocoptera)*.

**Figure 2 insects-14-00798-f002:**
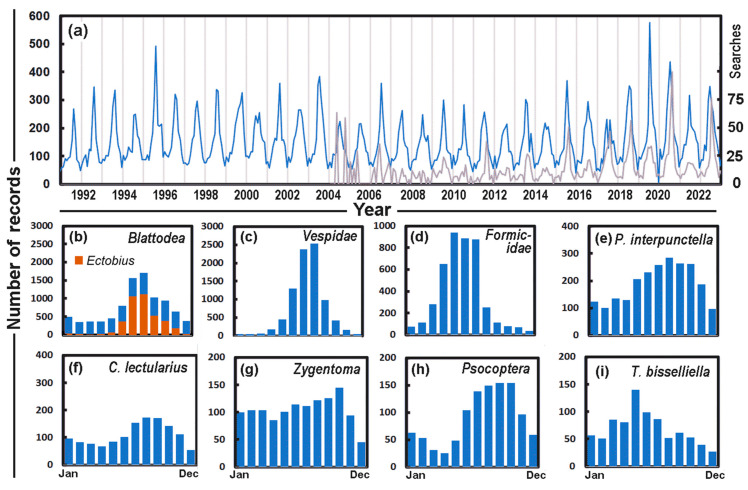
(**a**) The number of reports of *Vespidae* each month (blue) compared with the relative number of Google searches each month for “Wespe” in Switzerland (grey scale on right). Monthly numbers during 1991–2022 for (**b**) *Blattodea* with *Ectobius* denoted in brown, (**c**) *Vespidae*, (**d**) *Formicidae*, (**e**) *P. interpunctella*, (**f**) *C. lectularius*, (**g**) *Zygentoma* (**h**) *Psocoptera*, and (**i**) *T. bisselliella*.

**Figure 3 insects-14-00798-f003:**
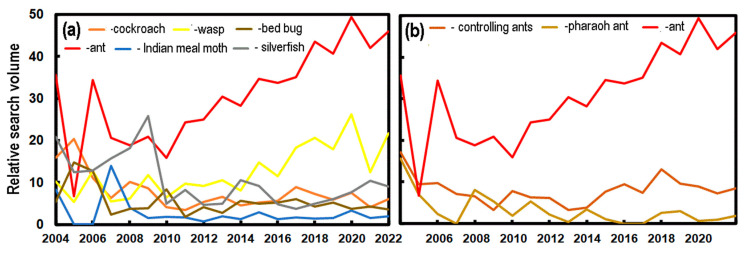
(**a**) The number of Google searches in Switzerland (normalised to a monthly maximum of 100) for: “Schabe” (cockroach), “Wespe” (wasp), “Ameise” (ant), “Dörrobstmotte” (Indian meal moth), “Bettwanze” (bed bug) and “Silberfischchen” (silverfish). (**b**) Google search volume for “Ameise” (ant), “Pharaoameise” (pharaoh ant) and “Ameisen bekämpfen” (controlling ants).

**Figure 4 insects-14-00798-f004:**
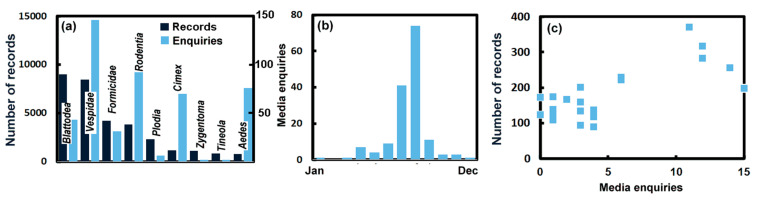
(**a**) The number of UPAS records and media enquiries (scale to right) about *Blattodea*, *Vespidae*, *Formicidae*, *Rodentia* (*Rattus* sp. and *Mus* sp.), *P. interpunctella*, *C. lectularius*, *Zygentoma*, *T. bisselliella* and *Ae. Albopictus*. (**b**) The total number of monthly media enquiries 1990–2022. (**c**) The number of UPAS records of the *Vespidae* each in the months June–August of each year as a function of the annual number of media enquiries for July–August of the years 2000–2022.

**Figure 5 insects-14-00798-f005:**
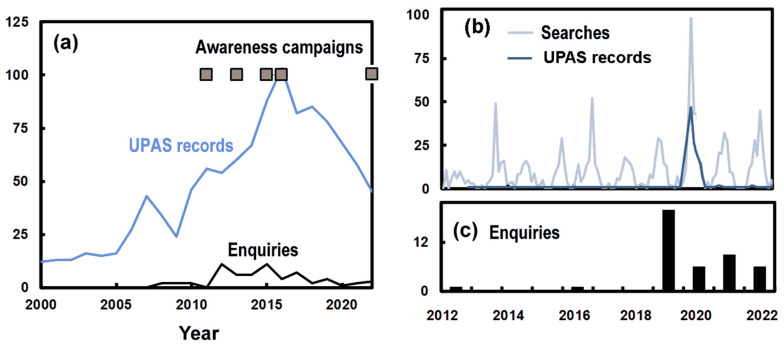
(**a**) The monthly number of records and media enquiries concerning *C. lectularius*, along with the years with media campaigns. (**b**) The monthly records of enquiries regarding *Ae. albopictus* in the UPAS database compared with the number of searches for “Tigermücke” from Google Trends. (**c**) Histogram of the monthly number of enquiries from the media about *Ae. albopictus*.

**Figure 6 insects-14-00798-f006:**
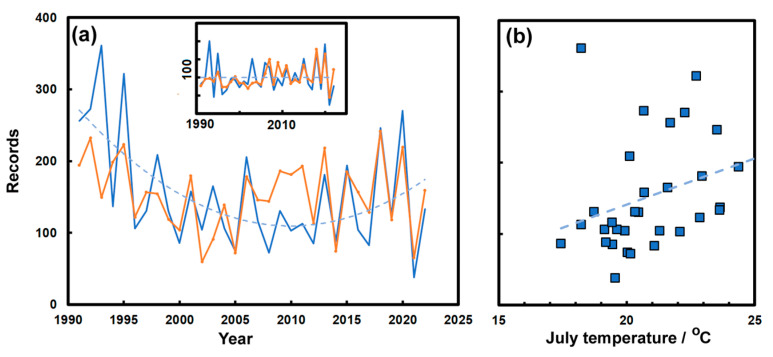
(**a**) Records of *Vespidae* from July and August over the years 1991–2022 (blue line) compared with those from the fit to Equation (1) (orange line). The dotted line shows the quadratic trend in the records. The inset shows the temperature polynomial fit once the trend has been removed, with the fitted line a baseline. (**b**) Records of *Vespidae* from July and August each year (1991–2022) as a function of the mean July temperature in Zurich 1991–2022 (blue line is a linear best fit).

**Figure 7 insects-14-00798-f007:**
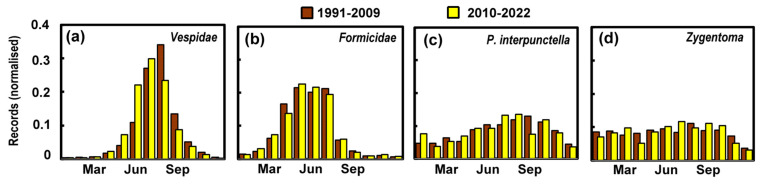
Normalised seasonal pattern of reports across the period 1991–2009 and 2010–2022 for (**a**) *Vespidae*, (**b**) *Formicidae*, (**c**) *Plodia interpunctella*, and (**d**) *Zygentoma*.

**Table 1 insects-14-00798-t001:** The number of records in the UPAS database for years before and after the COVID-19 pandemic. Note: 2018/19 and 2020/21 are averages.

	2018/19	2020/21	2022
*Rattus*	50.5	87	93
*Mus*	114.5	144.5	116
*Blattodea*	146	129.5	121
*Vespidae*	310.5	319	282
*Formicidae*	119.5	219.5	204
*Cimex*	81.5	63	45
*Plodia*	47.5	43.5	42
*Zygentoma*	34.5	49	45
*Psocoptera*	29.5	20	8
*Anthrenus*	32.5	30.5	36
*Attagenus*	35.5	39.5	34
*Stegobium*	18	24	15
*Columbidae*	5	22.5	27
*Halyomorpha*	20.5	4.5	0
*Monomorium*	12.5	12.5	4

## Data Availability

Data are available upon reasonable request to the UPAS.

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
