# Peer review of "Urban Pest Abundance and Public Enquiries in Zurich 1991–2022"

_insects, 2023, doi:10.3390/insects14100798_

Round 1

Reviewer 1 Report

This is an interesting paper on a long-term survey study on urban pests over a period of ~30 years in Switzerland. The paper was well-written and can be accepted with minor revisions. I have the following comments that I would like the authors to consider when revising the manuscript:

1. Please provide the authority name when a species name is mentioned for the first time. After that, the species name need not be spelled in full anymore unless it is used at the beginning of a sentence (eg. line 30, 33, 243)

2. Make sure all scientific names are italicized (eg. line 57, line 113).

3. Line 105, add "than" after more.

4. Please provide 0 before period - instead of using .0001, use 0.0001 (line 203). Please revise throughout the manuscript.

Author Response

Reviewer 1:

Comments and Suggestions for Authors

This is an interesting paper on a long-term survey study on urban pests over a period of ~30 years in Switzerland. The paper was well-written and can be accepted with minor revisions. I have the following comments that I would like the authors to consider when revising the manuscript:

1. Please provide the authority name when a species name is mentioned for the first time. After that, the species name need not be spelled in full anymore unless it is used at the beginning of a sentence (eg. line 30, 33, 243)

TOTALLY AGREE. HOWEVER AS THE ABSTRACT IS READ STAND-ALONE WE LEFT FULL SPELLING, BUT DID NOT ADD AUTHORITY TILL THE SPECIES IS ECOUNTERRED IN THE TEXT AS IT IS NOT USEUALTO REFERENCE ABSTRACTS. WE DID NOT GIVE AUTHORITIES FOR ORDERS OR GENERA, ONLY SPECIAS BINOMIALS WHEN THEY ARE FIRT INTRIDUCED . 
WE SPELL IN FULL VESPUKA AND VESPA IN OER TO BE CAR ON THE DISCTCTIOS BETWEEN THESE VESIDAE

2. Make sure all scientific names are italicized (eg. line 57, line 113).
THANK YOU FOR SEEING THOSE. 

3. Line 105, add "than" after more.
THANK YOU FOR SEEING THOSE. 

4. Please provide 0 before period - instead of using .0001, use 0.0001 (line 203). Please revise throughout the manuscript.
WE HAVE ADDED THESE IN ACCORDANCE WTH MDPI STYLE. HOWEVER WE INITIALLY ADOPTED THIS BECAUSE THE APA RECOMMENDS THAT THE LEADING ZEROES ARE OMITTED FROM PROBABLITIES.  

Reviewer 2 Report

Manuscript: Urban pest abundance and public enquiries in Zurich 1991-2022

Authors: Peter Brimblecombe, Gabi Müller, Marcus Schmidt, Werner Tischhauser, Isabelle Landau, Pascal Querner 

The authors present an interesting study in which they use data that has been gathered by UPAS and analyze it to detect patterns of change in pest trends according to various parameters, which include COVID-19, public and media submissions and inquiries, species, and methods of pest control. This work is unique in that it examines data that is submitted to an organization and analyzes it to observe and detect trends in pest activity. There is a lot of value in this type of information, and it often gets overlooked and not analyzed. This study is unique in the sense that it examines this type of data and looks at various parameters for patterns.

I have a couple of overarching concerns, which are as follows:

1.       The data being analyzed are inquiries that are submitted by various people; Media, pest control and the public. It is stated that within these inquiries submitted by the public things can be misidentified if not sent in to be verified by a member of UPAS (L101-104, L195-197, L282-285). It is concerning that the data being analyzed and thus drawing conclusions from is not accurate. No formal verification of the data was performed.

2.       The conclusions drawn from the data are speculation only. It could be true that there are patterns and trends due to COVID-19, changes in methods of pest control and seasonal changes. However, in order to show this, official documentation needs to be gathered to arrive at this conclusion. It was mentioned that surveys are difficult to do, because of the low return, but they must be done in order to validate the statements and claims within the study(L351-353). Surveys can be done to ask pest control companies what type of control methods are being used for specific pests, to arrive at the conclusion that it is because effective gel baits are being used (L18-21,L164-165, L199-200). Surveying the public to ask them about their experiences during COVID-19 in terms of pest control or activity. Were they noticing more pests around their home because they were home more (L445-446)? Or do they tend to search for certain pests online or submit inquiries more after awareness campaigns like stated in L344-437 and L301-304?

Additional Comments:

·         There is confusion in the results sections as to what data is being analyzed. It was difficult to follow if only the publics inputs were being analyzed, or if it was public, pest control companies and media. It is stated L108-110 that the data can be separated out, but in the results, it is not stated for each subsection of what data is being used.

·         A lot of the information in the results section would be better served to be moved to a discussion section or to the conclusion section. It made the results hard to follow, by stating what the findings were and then trying to explain them immediately.

·         The introduction gives a lot of information that may not be needed, such as the lengthy background of UPAS. Condensing the introduction to concentrate on what the UPAS is and what is does would be beneficial.

Additional comments presented by line:

L86-88 ending after “nuisance”, I am not sure this needs to be stated or how that strengthens the methods.

Figure 3.: In the legend putting the English term in parenthesis or replacing the German word with the English term would reduce confusion.

Figure 4a: The legend is confusing. “Enquiries x100”, for example, does this mean that Rodentia had about 1,000,000 enquiries?

Figure 4a: Consider moving the family names to the x-axis.

L375 Should be “3.5 Climate effects” – 3.4 section is present twice.

Figure 5a: It is confusing what the authors are presenting by this visual.

There are minor grammatical errors that should be addressed. 

Author Response

reviewer 2:

THE AUTHORS ARE HAPPY TO HAVE THE COMMENTS WHICH HAVE HEPED IMPROVE THE STUDY

Comments and Suggestions for Authors

The authors present an interesting study in which they use data that has been gathered by UPAS and analyze it to detect patterns of change in pest trends according to various parameters, which include COVID-19, public and media submissions and inquiries, species, and methods of pest control. This work is unique in that it examines data that is submitted to an organization and analyzes it to observe and detect trends in pest activity. There is a lot of value in this type of information, and it often gets overlooked and not analyzed. This study is unique in the sense that it examines this type of data and looks at various parameters for patterns.

I have a couple of overarching concerns, which are as follows:

1.     The data being analyzed are inquiries that are submitted by various people; Media, pest control and the public. It is stated that within these inquiries submitted by the public things can be misidentified if not sent in to be verified by a member of UPAS (L101-104, L195-197, L282-285). It is concerning that the data being analyzed and thus drawing conclusions from is not accurate. No formal verification of the data was performed.

THIS POINT IS CORRECT. HOWEVER, WE HAVE BEEN CAREFUL TO USE THE WORD "REPORT" OR "REPORTING" RATHER THAN "CATCH" OF INSECTS. 
HOWEVER, IT IS IMPORTANT NOT TO HIDE BEHIND THIS LINGUISTIC DISCTINCTION OF "REPORT" OR "REPORTING", SO WE HAVE EXPANDED THE DISCUSSION OF THE ACCURACY OF SPECIES DETERMINATION IN THE NEW SECTION 3.7 WHERE WE DISCUSS THE BIAS ASSOCIATED WITH SELF-REPORTED DATA AND IN PARTICULAR NOW DISCUSS HOW SELF-REPORTING BIAS IS HANDLED IN OTHER FIELDS

THIS IS A FAMILIAR PROBLEM IN SELF-REPORTING OF HEALTH STATUS TO QUOTE:  "When you’re asking people about their health habits, there is room for bias, especially when that person may be concerned about how their answers will be used. There is also room for error in personal responses—how accurately do you think you could represent the amount of exercise you get in the past month? But if you overlook this type of data, you lose out on a critical dataset in your arsenal that can give you additional insights and layers of understanding when it comes to your population’s health. "

2.     The conclusions drawn from the data are speculation only. It could be true that there are patterns and trends due to COVID-19, changes in methods of pest control and seasonal changes. However, in order to show this, official documentation needs to be gathered to arrive at this conclusion. It was mentioned that surveys are difficult to do, because of the low return, but they must be done in order to validate the statements and claims within the study(L351-353). Surveys can be done to ask pest control companies what type of control methods are being used for specific pests, to arrive at the conclusion that it is because effective gel baits are being used (L18-21, L164-165, L199-200). Surveying the public to ask them about their experiences during COVID-19 in terms of pest control or activity. Were they noticing more pests around their home because they were home more (L445-446)? Or do they tend to search for certain pests online or submit inquiries more after awareness campaigns like stated in L344-437 and L301-304?

TRUE. IN A SENSE THE INTERPRETATION OF MOST INSECT TRAPPING DATA ARE SPECULATIVE AS THEY RELY ON ON STATISTICAL ESTMATES OF PROBABILITY OR CORRELATION, WHICH CANNOT IN ITSELF CONFIRM CAUSALITY. THIS IS NOW SPECIFICALLY MENTIONED AS. "although statistical analysis was used to assess the probability of changes in the sug-gested by the data, at offers less guidance on the causes of change. "  

WE WORRY THAT SURVEYING THE PUBLIC THREE YEARS ON FROM COVID MIGHT NOT REALLY GIVE A RELIABLE GOOD REFLECTION OF THEIR EXPERIENCE (in epidemiology this bias is seen as a recall issue). HOWEVER, we PLACE THESE IDEAS AS FURTHER RESEARCH IN THE CONCLUSIONS AS:  "We have still to undertake surveys of the public experience of urban pests and explore the ways in which analysis of the data contributes to better management of urban pests. "

DATA FROM PEST CONTROL COMPANIES CAN BE DIFFICULT TO OBTAIN, BUT WE HAVE NOW MENTIONED THAT WE THEY WORK IN PUBLIC BUILDINGS THEY ARE REQUIRED TO PROVIDE SPECIMENS. "Unfortunately, it is difficult to get returns to these surveys because of an understanda-ble commencal sensitivity, so only six replies were otaiined from more than 50 companies in the Swiss Pest Control Association. This reflected a reluctance to reveal methods or properties treated, but now pest control work in public buildings required to submit specimens of insects caught, so there will be additional data in future. "

Surveys can be done to ask pest control companies what type of control methods are being used for specific pests, to arrive at the conclusion that it is because effective gel baits are being used (L18-21, L164-165, L199-200). 

WE COVER ONLY GEL BAITS IN THIS MS, BUT ARE PREPARING A PAPER ON Aedes albopictus WHERE METHODS USED IN PEST CONTROL WILL BE COVERED IN MORE DETAIL.

Additional Comments:

·         There is confusion in the results sections as to what data is being analyzed. It was difficult to follow if only the publics inputs were being analyzed, or if it was public, pest control companies and media. It is stated L108-110 that the data can be separated out, but in the results, it is not stated for each subsection of what data is being used.

THE CASES WHERE ELEMENTS OF THE DATASET WERE USED ARE NOW DESRCRIBED IN MORE DETAIL AND THIS ISSUE IS NOW IS MENTIONED AS "We were able to separate out media requests for information about insects and sub-missions from pest control companies. This was useful to explore journalistic activities and new regulations for reporting pest eradication efforts in public buildings. " 

UNDERSTAND, BUT SOME PEOPLE PREFER HAVING THE DISCUSSION IMMEDIATELY FOLLOW EACH SET OF RESULTS SO THE FIGURES BEING DISCUSSED ARE ADJACENT ThE TEXT - TO MEET THE REFEREE'S CONCERNS  WE HAVE SPLIT MATERIAL, WHICH CAN BE READILY SEPARATED INTO A FINAL SECTION OF THE RELABELED THE "RESULTS" SECTION NOW AS "RESULTS AND DISCUSSION". THE PAPER STRUCTUREE HAS BEEN BY THE MDPI GUIDANCE FOR SECTION HEADINGS AND CONTENT. 

·         A lot of the information in the results section would be better served to be moved to a discussion section or to the conclusion section. It made the results hard to follow, by stating what the findings were and then trying to explain them immediately.

SEE COMMENTS ABOVE

·         The introduction gives a lot of information that may not be needed, such as the lengthy background of UPAS. Condensing the introduction to concentrate on what the UPAS is and what is does would be beneficial.

AGREE THIS HAS NOW BEEN SHORTENED. 

Additional comments presented by line:

L86-88 ending after “nuisance”, I am not sure this needs to be stated or how that strengthens the methods.
AGEEE SO SHORTENED TO "Data collection by the UPAS was not primarily designed to support its official ac-tivities, with reports and enquiries to the UPAS are stored in Microsoft Dynamics 365 and have similarities with the collection of data within medical entomology departments [4], historic houses [5,6] or museums [7]. Often these use Microsoft Excel to handle data, and although such collections of data are valuable in research, they need to be tuned to the needs of studies [8]."

Figure 3.: In the legend putting the English term in parenthesis or replacing the German word with the English term would reduce confusion.
DONE AS ENGLISH ADDITIONS TO LEGEND

Figure 4a: The legend is confusing. “Enquiries x100”, for example, does this mean that Rodentia had about 1,000,000 enquiries?
AGREE IT WAS CONFUSING SO THE RIGHT HAND AXIS HAS BEEN RELABELLED

Figure 4a: Consider moving the family names to the x-axis.
THE LENGTH OF NAMES MADE THIS VERY DIFFICULT SO LEFT VERTICAL - THE SAME PROBLEM OCCURS IN FIGURE 1a

L375 Should be “3.5 Climate effects” – 3.4 section is present twice.
CHANGES - THANK YOU!

Figure 5a: It is confusing what the authors are presenting by this visual.

AGREE THIS IS NOW EXPLAINED IN MORE DETAIL AS "Public awareness is understandably enhanced by information campaigns, most notably for issue with C. lectularius (bed bug). The UPAS produced press releases about prevention before the start of the summer holidays in the years 2011, 2013, 2015, 2016 and 2022 (Figure 5a), how to avoid bringing bed bugs home when returning from the holidays. These press releases aligned with a growing numbers of reports to the UPAS (peaking in 2016), and continued media attention in previous years."

There are minor grammatical errors that should be addressed. 
THANK YOU WE HAVE MADE CHANGES TO THE GRAMMAR

Round 2

Reviewer 2 Report

I appreciate addressing my concerns by responding to them and making changes to make the manuscript stronger. I have a few suggestions:

1. Figure 3: The legend looks too cluttered. I would consider removing the German term on the legend, but leaving in the figure description.

2. Figure 4: "Rodentia" in the figure is blocking the dark blue bar. Can it be moved slightly higher on the light blue bar, without blocking the top, like it is the dark blue bar currently.

Can the x-axis on Fig. 4 (c) be shifted so there are not points on the border of the graph. For example, have the x-axis be -2 to 17? Only have increments of 5 labelled on the graph like it currently is.

Fig 5(a): same comment as above. Shift the x-axis so there is not a point on the graph border.

There are still some minor English and grammatical errors throughout that need to be fixed.

Author Response

We have changed Fig 3 as suggested, but needs new caption:   Figure 3. (a) The number of Google searches in Switzerland (normalised to a monthly maximum of 100) for: “Schabe” (cockroach), “Wespe” (wasp), “Ameise” (ant), “Dörrobstmotte” (Indian meal moth) , “Bettwanze” (bed bug) and “Silberfischchen” (silverfsh). (b) Google search volume for “Ameise” (ant), “Pharaoameise” (pharoah ant) and "Ameisen bekämpfen" (controlling ants).    We have changed Fig 4 as suggested   We have otlined the points on the axes in white so they are more distinctive    We have changed Fig 5 as surgested, but as with 4c have added a white border to the poin on the axis.
